# Conditional generation of medical time series for extrapolation to underrepresented populations

**Simon Bing**[1,2]*, **Andrea Dittadi**[3], **Stefan Bauer**[4,5,6‡], **Patrick Schwab**[6‡]

**1** ETH Zürich, Zürich, Switzerland, **2** Max Planck Institute for Intelligent Systems, Tübingen, Germany, **3** Technical University of Denmark, Copenhagen, Denmark, **4** KTH Stockholm, Stockholm, Sweden, **5** CIFAR Azrieli Global Scholar, Toronto, Canada, **6** GlaxoSmithKline, Artificial Intelligence & Machine Learning, Zug, Switzerland

‡ These authors are joint senior authors on this work.
* simon.bing95@gmail.com

**Data Availability Statement:** The utilized MIMIC-III data set (https://physionet.org/content/mimiciii/1. 4/) is publicly available to researchers after having completed a course to certify their capability to

## Abstract

The widespread adoption of electronic health records (EHRs) and subsequent increased availability of longitudinal healthcare data has led to significant advances in our understanding of health and disease with direct and immediate impact on the development of new diagnostics and therapeutic treatment options. However, access to EHRs is often restricted due to their perceived sensitive nature and associated legal concerns, and the cohorts therein typically are those seen at a specific hospital or network of hospitals and therefore not representative of the wider population of patients. Here, we present HealthGen, a new approach for the conditional generation of synthetic EHRs that maintains an accurate representation of real patient characteristics, temporal information and missingness patterns. We demonstrate experimentally that HealthGen generates synthetic cohorts that are significantly more faithful to real patient EHRs than the current state-of-the-art, and that augmenting real data sets with conditionally generated cohorts of underrepresented subpopulations of patients can significantly enhance the generalisability of models derived from these data sets to different patient populations. Synthetic conditionally generated EHRs could help increase the accessibility of longitudinal healthcare data sets and improve the generalisability of inferences made from these data sets to underrepresented populations.

## Author summary

Electronic health record (EHR) data sets are essential for developing machine learning (ML) based therapeutic and diagnostic tools. Developing such data-driven models requires large and diverse amounts of medical data, but access to the necessary data sets is often not given in practice. Even when access is provided, the data usually stems from a single source, resulting in models that preform well for the patient groups from this limited source, but not on the diverse, general population. Here, we introduce a new method to generate synthetic EHR patient data, helping to overcome the issues of data access and

handle sensitive patient data. Access to the data may be requested at https://mimic.mit.edu/.

**Funding:** The authors received no specific funding for this work.

**Competing interests:** I have read the journal's policy and the authors of this manuscript have the following competing interests: St.B. and P.S. are employees and shareholders of GSK. The remaining authors declare no competing interests.

patient representation. The data that our method generates shares the characteristics of real patient data, allowing the developers of downstream ML models to use this data freely during development. With our model, we can directly control the composition of patient cohorts, in terms of demographic variables such as age, sex and ethnicity. We can therefore generate more representative data sets, which lead to more fair downstream models and ultimately a more fair treatment of underrepresented populations.

## 1 Introduction

The broad use of electronic health records (EHRs) has lead to a significant increase in the availability of longitudinal health care data. As a consequence, our understanding of health and disease has deepened, allowing for the development of diagnostic and therapeutic approaches directly derived from EHR patient data. Models that utilize rich healthcare time series data derived from clinical practice could enable a variety of use cases in personalised medicine, as evidenced by the numerous recent efforts in this area [1–4]. However, the development of these novel diagnostic and therapeutic tools is often hampered by the lack of access to actionable patient data [5].

Even after being deidentified, EHR data is perceived as highly sensitive and clinical institutions raise legal and privacy concerns over the sharing of patient data they may have access to [6]. Furthermore, even if data is made public, it often originates from a single institution only [7–9], resulting in a data set that may not be representative for more general patient populations. Basing machine learning models on single site data sets only risks overfitting to a cohort of patients that is biased towards the population seen at one clinic or hospital, and renders their use for general applications across heterogeneous patient populations uninformative at best and harmful at worst [10, 11].

Putting aside the issue of non-representative patient cohorts, the development of accurate machine learning-based models for healthcare is further impeded by the imbalance in magnitude of the available data compared to other domains. While fields such as computer vision or language modelling have made significant advances, thanks in part to access to large-scale training data sets like ImageNet [12] or large text corpora derived from the World Wide Web, there do not yet exist any comparable data repositories for machine learning in healthcare that may spur innovation at similar pace. Practical problems may also arise during model development due to a lack of training samples for specific, rare patient conditions. If one wishes to study a model's behaviour given data with certain properties, such as only patients with a certain pre-existing condition, medical data sets may often be too small to representatively cover such populations.

One potentially attractive approach to address the aforementioned issues would be to generate realistic, synthetic training data for machine learning models. Given access to an underlying distribution that approximates that of the real data, paired with the capability to sample from it, one could theoretically synthesize data sets of any desired size. The generated synthetic patient data can be used for assessing [13–15] or even improving machine learning healthcare based software e.g. for liver lesions classification [16]. If the generative model of the data were to also have the capacity to generate samples conditioned on factors that may be freely chosen, such as for example pre-existing conditions, data sets with the exact properties required for a specified task could be generated as well. Previous reports suggest that such synthetically generated data sets may furthermore be shared with a significantly lower risk of exposing private information [17].

Beyond generating synthetic data to address issues surrounding fairness and bias mitigation, other complementary approaches have been studied in the literature. These include methods to transfer learned knowledge from one data set to another [18, 19], casting the collection of training data as an optimization problem with an objective function directly linked to population-level goals [20], as well as meta-learning approaches that generalize to a new task with relatively few samples [21]. Considering confounding factors is another important point when addressing issues surrounding fairness and bias in machine learning applications for medicine [22, 23]. While these and other approaches to mitigating bias in medical data sets [24] show promise to aid in the development of more fair clinical machine learning tools, we propose a complementary approach based on synthesizing medical data sets to do so. Specifically, our method is characterized by its capability to conditionally generate data, thereby directly modelling the effect of otherwise confounding variables.

Developing models with synthetic data is already widely applied in machine learning research. In Reinforcement Learning for example, it is the de-facto standard to train models in simulation, in order to have high-fidelity control over the environment [25, 26], or simply because experiments in the real world would be to costly, unethical or dangerous to conduct. Some previous work even suggests that models trained on synthetic data could outperform those derived from real data sets [27]. The gap between real and synthetic data is rapidly closing in fields like facial recognition in computer vision, as has for example recently been demonstrated by Wood et al. [28].

Classical approaches to generating medical time series data exist, but they fall short of the requirements that modern data-driven models require for their input. Some works employ hand crafted generation mechanisms followed by expensive post-hoc rectifications by clinical professionals [17], while others rely on mathematical models of biological subsystems such as the cardiovascular system [29, 30], which require a detailed physiological understanding of the system to be modelled. When the output data stems from multiple, interconnected subsystems whose global dynamics are too complex to model with ordinary differential equations and the size of the required data set is too large to tediously correct unrealistic samples by experts, these approaches may be difficult to utilize.

A natural approach to learning complex relationships from data is to move away from hand-crafted generative models and utilize machine learning methods. While a plethora of powerful generative models for medical imaging data generation have been brought forward in recent years [31–34], relatively little research has been reported on generating synthetic medical time series data [5, 35–37]. Moreover, the generation and evaluation of synthetic patient data [38] is often challenging due to the high prevalence of missing measurement values in the original medical data sets [39–42].

To address these issues, we present HealthGen, a new approach to conditionally generate EHRs that accurately represent real measured patient characteristics, including time series of clinical observations and missingness patterns. In this work, we demonstrate that the patient cohorts generated by our model are significantly better-aligned to realistic data compared to various state-of-the-art approaches for medical time series generation. We demonstrate that our model outperforms previous approaches due to its explicit development for real clinical time series data, resulting in modelling not only the dynamics of the clinical covariates, but also their patterns of missingness which have been shown to be potentially highly informative in medical settings [43]. We show that our model's capability to synthesize specific patient subpopulations by means of conditioning on their demographic descriptors allows us to generate synthetic data sets which exhibit more fair downstream behavior between patient subgroups, than competing approaches. Moreover, we demonstrate that by conditionally generating patient samples of underrepresented subpopulations and augmenting real data sets to equally

represent each patient group, we can significantly boost the fairness (In this work, we measure fairness in terms of generalization performance to underrepresented populations.) of downstream models derived from the data. Furthermore, we evaluate the quality and usefulness of the data we generate using a downstream task that represents a realistic clinical use-case—allowing us to compare our model against competing approaches in a setting that is relevant for clinical impact.

Our main contributions are:

- We introduce HealthGen, a new machine-learning method to conditionally generate realistic EHR data, including patient characteristics, the temporal evolution of clinical observations and their associated missingness patterns over time.

- We experimentally show that our method outperforms current state-of-the-art models for medical time series generation in synthetically generating patient cohorts that are faithful to real patient data.

- We demonstrate the high fidelity of control over synthetic cohort composition that our model provides by means of its conditional generation capability, by generating more diverse synthetic data sets than competing approaches, which ultimately leads to a more fair representation of different patient populations.

- We show that by augmenting real data with conditionally generated samples of underrepresented populations, the models derived from these data sets exhibit significantly more fair behaviour than those derived from unaltered real data.

- We perform a comprehensive computational evaluation in realistic clinical use cases to evaluate the comparative performance of HealthGen against various state-of-the-art generative time-series modelling approaches.

## 2 Results

### 2.1 Overview

For this study, we consider the MIMIC-III data set [8], which consists of EHRs containing time series of measurements of patients that spent time in the intensive care units (ICUs) of the Beth Israel Deaconess Medical Center in Boston, Massachusetts, USA. Additionally, each patient is described by static variables such as their age, ethnicity, insurance type and sex. The time series of a given patient is labelled to indicate whether or not one of the following clinical interventions was performed: mechanical ventilation, vasopressor administration, colloid bolus administration, crystalloid bolus administration or non-invasive ventilation.

After extracting the cohort of patients from the data base, we split them into training (70%), validation (15%) and test (15%) sets, stratified by their binary intervention labels. The generative models are trained on the real training data $\mathcal{D}_{\text{train}} = \left\{ \mathbf{x}_{1:T}^n, \mathbf{m}_{1:T}^n, \mathbf{s}^n, \mathbf{y}^n \right\}_{n=1}^{N_{\text{train}}}$, where $\mathbf{x}_{1:T}$ denotes the time series of covariates, $\mathbf{m}_{1:T}$ the time series of binary masks indicating where values are missing, $\mathbf{s}$ the static patient variables and $\mathbf{y}$ a patient's set of labels for the respective clinical interventions. We include the missingness information $\mathbf{m}_{1:T}$ explicitly, as previous work has shown that patterns of missing values are highly informative [41] and especially in the medical setting including them is preferential to imputation [43].

To evaluate and compare generative models, we first train a downstream time series classification model developed for medical data [43] on the data sets synthesized by each model. This classifier that has been trained with synthetic data is evaluated on the held-out real test data, and the resulting AUROC score (Area Under the Receiver Operating Characteristic curve) is

compared with the AUROC score of the same classifier trained on the real data. The final measure for how faithful a given generated data set is to the real data is the difference between the evaluation score of the synthetic data and that of the real data. Details on the experimental pipeline can be found in Section 4.

In our first experiment, we generate synthetic patient cohorts that are faithful to the real data in terms of their demographics, i.e. they contain the same number of patients as the original data under the same distribution of static variables. Repeating this synthetic cohort generation for all of the available clinical intervention labels, we compare the performance of our model to baseline models for generating clinical time series data. As baselines, we consider Stochastic Recurrent Neural Networks (SRNN) [44] and KalmanVAE (KVAE) [45], both based on variational autoencoders (VAE) [46], and TimeGAN [47], based on generative adversarial networks (GAN) [48]. We then present results of an extension of the previous experiment demonstrating our proposed approach's conditional generation capability, where we generate patient cohorts with static variable distributions that differ from the real data, and investigate the fairness of models derived from the resulting synthetic data. In an additional experiment, we identify real data settings where some subpopulations of patients have a significantly lower classification score than other patients. Using the the conditional generation capability of our model, we augment the real data with synthetic samples of minority groups and test if this augmentation leads to an increase in the downstream classification score of the previously underrepresented populations.

## 2.2 Generating synthetic patient cohorts

In the first experiment of this work, we investigate our model's capability to generate synthetic data that is faithful to the real data and useful in downstream tasks. To study the generative performance of our model and compare it against competing approaches for medical time series generation, we employ the experimental framework described in detail in the Methods section.

Here, we train each generative model conditioned on the real labels $\mathbf{y}$ and then generate a synthetic data set $\hat{\mathcal{D}}$, where the generation is again conditioned on the label of the considered task, to guarantee that the synthetic data shares the same statistics in terms of split between positive and negative labels as the real data. While we could in practice generate as much data as we wish, we synthesize data sets containing the same number of patients as the real data for this experiment, to facilitate a fair comparison between the real patient cohorts and those that have been synthetically generated.

A downstream model for medical time series classification [43] is then trained on the synthetic training data $\hat{\mathcal{D}}_{\text{train}}$ of each generative model and evaluated on a held out test data set $\mathcal{D}_{\text{test}}$ consisting of real patients. We compare the performance of our model against the current state-of-the-art models for time series generation, across all five available classification tasks in the MIMIC-III data. The results of these experiments are summarized in Table 1. We provide

**Table 1. Comparison of AUROC scores for all predictive tasks between HealthGen and the baseline models.** The 95% confidence interval of the mean value is presented in parentheses and is estimated via bootstrapping with 30 samples.

|  | vent | vaso | colloid_bolus | crystalloid_bolus | niv |
|---|---|---|---|---|---|
| Real Data | 0.809 (0.807, 0.811) | 0.801 (0.799, 0.803) | 0.751 (0.741, 0.760) | 0.613 (0.609, 0.616) | 0.634 (0.632, 0.637) |
| HealthGen (Ours) | **0.769 (0.767, 0.772)** | **0.722 (0.718, 0.727)** | **0.664 (0.650, 0.678)** | **0.574 (0.571, 0.577)** | **0.567 (0.566, 0.569)** |
| SRNN | 0.639 (0.637, 0.641) | 0.693 (0.690, 0.695) | 0.661 (0.656, 0.666) | 0.562 (0.561, 0.564) | 0.553 (0.552, 0.555) |
| KVAE | 0.559 (0.549, 0.570) | 0.608 (0.589, 0.627) | 0.565 (0.544, 0.586) | 0.538 (0.531, 0.545) | 0.523 (0.517, 0.529) |
| TimeGAN | 0.558 (0.558, 0.558) | 0.703 (0.703, 0.704) | 0.552 (0.530, 0.573) | 0.545 (0.543, 0.548) | 0.530 (0.527, 0.533) |

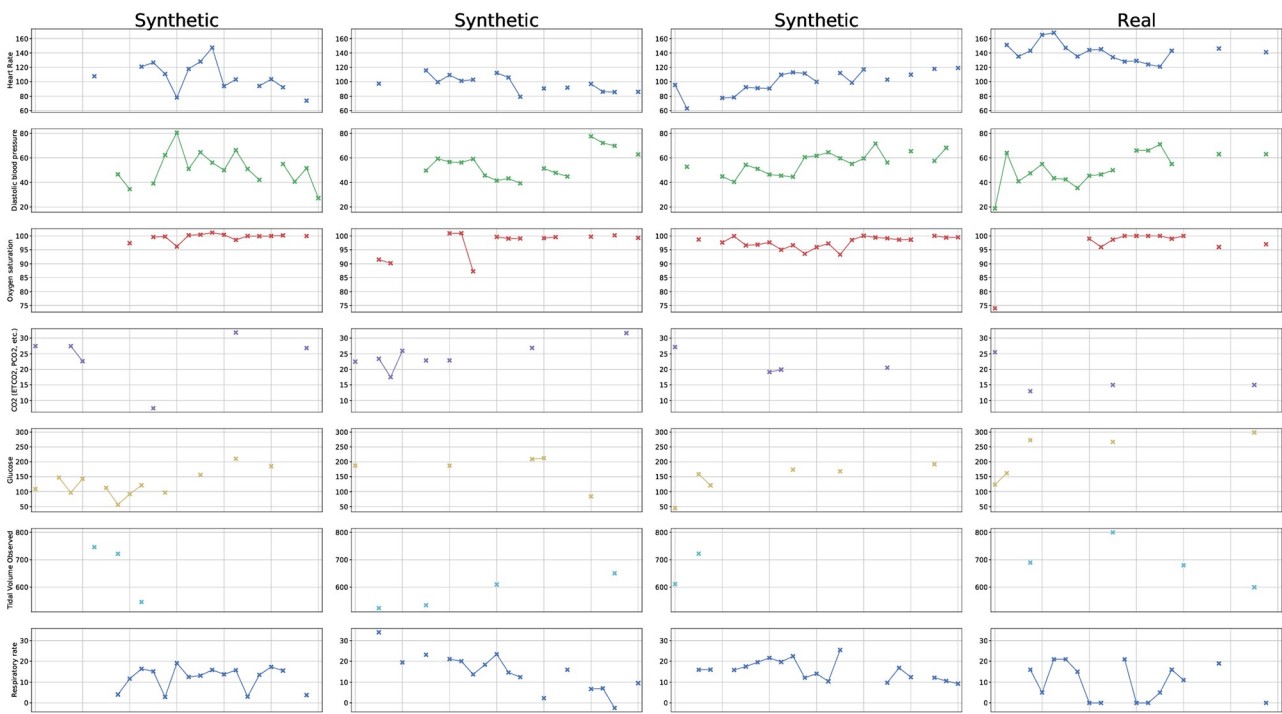

**Fig 1. Sample time series of synthetically generated patients, with the time series of one real patient for comparison.**

examples of synthetically generated patients' time series in Fig 1, next to a real patient's time series for comparison. Additional, more detailed visualizations are presented in S4 Appendix. While these visualizations show a qualitative similarity between synthetic and real time series, they further underscore the need for our utilized, more fine-grained quantitative evaluation of the quality of the generated data.

In all but one of the considered tasks, our approach significantly outperforms the state-of-the-art models in synthetically generating medical time series. The fact that our model's downstream classification score is higher then the baselines', across all experimental settings, suggests that the synthetic data generated by our model is more faithful to the real data and thus more useful for the development of downstream clinical time series prediction tasks than the cohorts generated by any of the competing architectures.

## 2.3 Conditional generation

From Table C in S3 Appendix we see that many demographic variables have examples of highly underrepresented classes, possibly leading to the classification performance for these subpopulations to be much lower than for the majority class. This shines a light on a real problem found in many clinical settings, especially when transferring between hospitals [3]. In preliminary experiments, we investigate the classification score on a per-group level for the real data (cf. Table E in S1 Appendix). While it does not occur for all static variables and classification tasks, we identify cases where there is a significant difference in the score of a given subpopulation, with respect to the other groups. This inter-group performance gap raises the question if our model's conditional generation capability can be leveraged to address these fairness issues.

In the preceding experiment, we do no utilize our model's capacity to conditionally generate synthetic patient cohorts, as we do not explicitly condition our model on the static variables

**s** during training or generation. Here, in addition to the label **y**, we condition on a static variable of interest allowing us to then conditionally generate an equal number of synthetic patient samples for each subgroup of this considered static variable. We investigate if conditionally generating patient cohorts provides a benefit in terms of fairness, as well as overall downstream performance. In this experiment, we study the performance on a per-group level, comparing not only to the baseline approaches, but to our model when unconditionally generating the data, as well. To enable a fair comparison, the overall number of generated patients is equal for each considered model. The results of this experiment for two exemplary settings are presented in Fig 2, with additional results reported in Fig A in S1 Appendix.

The results indicate that, for settings in which the real data exhibits performance differences between subpopulations, conditionally generating synthetic patient cohorts provides a significant benefit over unconditional generation. In terms of overall performance, and in nearly all subgroups, our model outperforms the baselines when conditionally generating data. The score of the fairness metric we utilize is also higher for our model, than for any of the competing approaches (cf. Table C in S1 Appendix). Our model is not only capable of generating synthetic "copies" of the real data in terms of the distribution of demographics, but we can generate data sets with a high degree of control over the composition of subpopulations, resulting in more diverse training sets for downstream tasks.

## 2.4 Real data augmentation

In the preceding experiment, we demonstrated that our model's conditional generation capability can be used to synthesize patient cohorts that yield more fair downstream classification models. The settings that emerge in which our model can provide a benefit are those where the real data displays an imbalance in the downstream performance between patient subpopulations. This gives rise to the question if our approach to conditionally generate synthetic data can also be useful for the setting when access to the real data is not restricted, but rather the given cohort does not fulfill the requirements for the development of downstream models. For these cases, we hypothesize that we can conditionally generate more examples of this previously underrepresented class, augment the real data with them and thereby boost the performance in the downstream classification task for this subpopulation.

One of the cases, where we identify a significant difference in the performance of the trained classifier for different subtypes of patients, is the `colloid_bolus` classification task when looking at different insurance types of patients. In Fig 3, we see that while the overall score on this task is fairly high, the underrepresented class of Government insured individuals has a significantly lower score than all other classes. The score of this class is even lower than 0.5, which would be obtained by randomly guessing which class a sample belongs to.

To investigate if our model can improve the performance for such an underrepresented group, we conditionally generate additional samples of each underrepresented class and augment the real training data with these until all insurance types are fairly represented by the same number of samples. Since the baselines cannot generate data conditioned on static variables, we have them unconditionally generate the same number of overall samples that our model augments the real data with. We then compare the results of the downstream classifier trained on data sets augmented by synthetic data of the respective generative models to the classifier trained on the real data.

As we see in Fig 3(a), our model can indeed increase the performance of previously underrepresented groups. Our model significantly boosts the predictive score of the Government class, without sacrificing the performance of any other subpopulation. This result is remarkable, since the number of positive samples of Government insured individuals is exceedingly

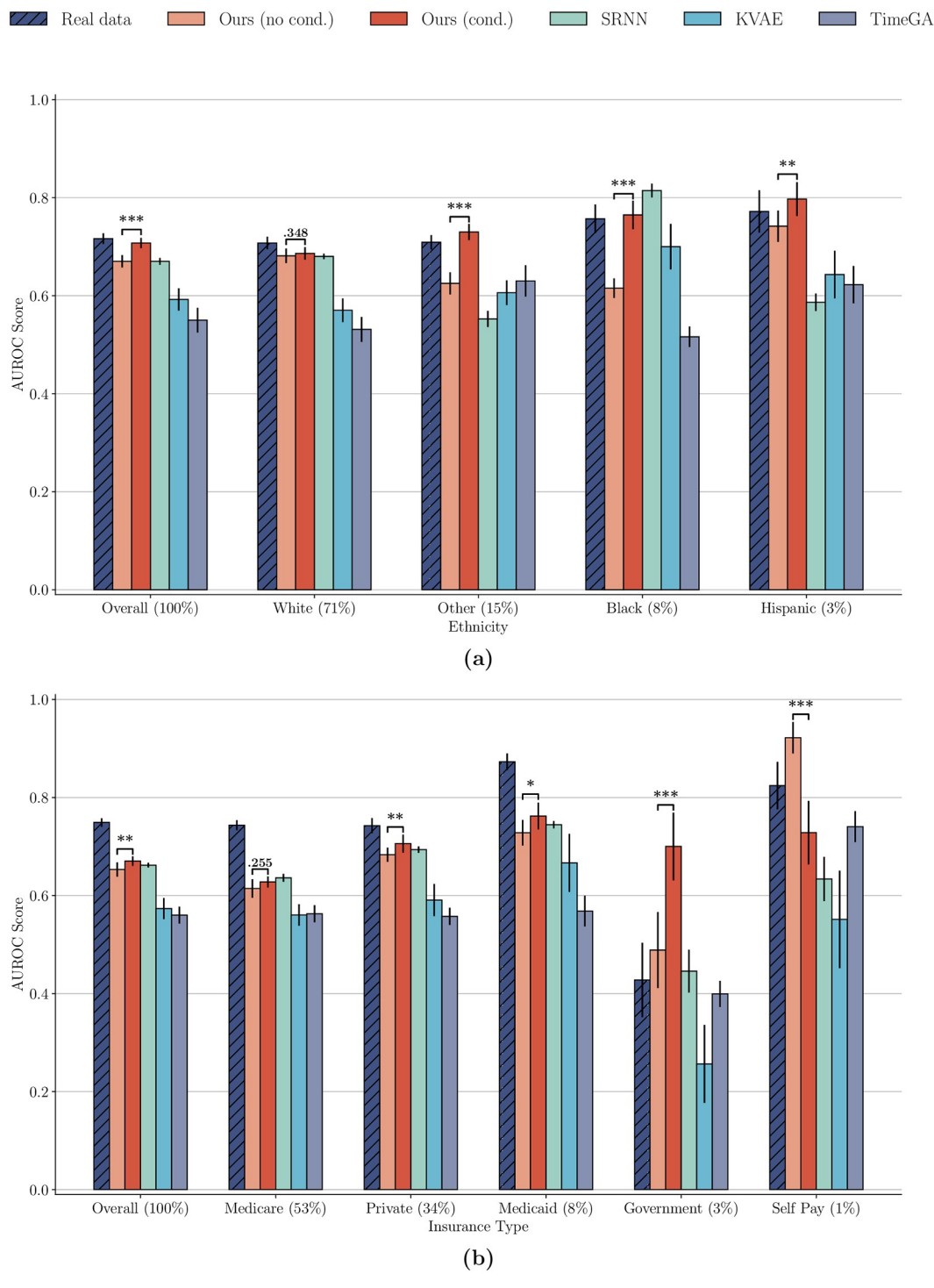

**Fig 2. Comparison of AUROC score between our model when conditionally and unconditionally generating synthetic data with baselines.** We show results of the `colloid_bolus` task for different ethnicity groups (a) as wells as the range of insurance types (b). Note that the Asian subpopulation is not among the ethnicity groups, as there are no positive validation samples among this group for the considered task, thereby prohibiting the calculation of the AUROC score. Conditionally generating patient cohorts is favourable to unconditional generation overall, and for almost all subgroups, allowing for the generation of more representative and therefore fair synthetic data sets. Significance levels between groups of interest are shown with brackets, where * corresponds to p < 0.05, ** to p < 0.01 and *** to p < 0.001.

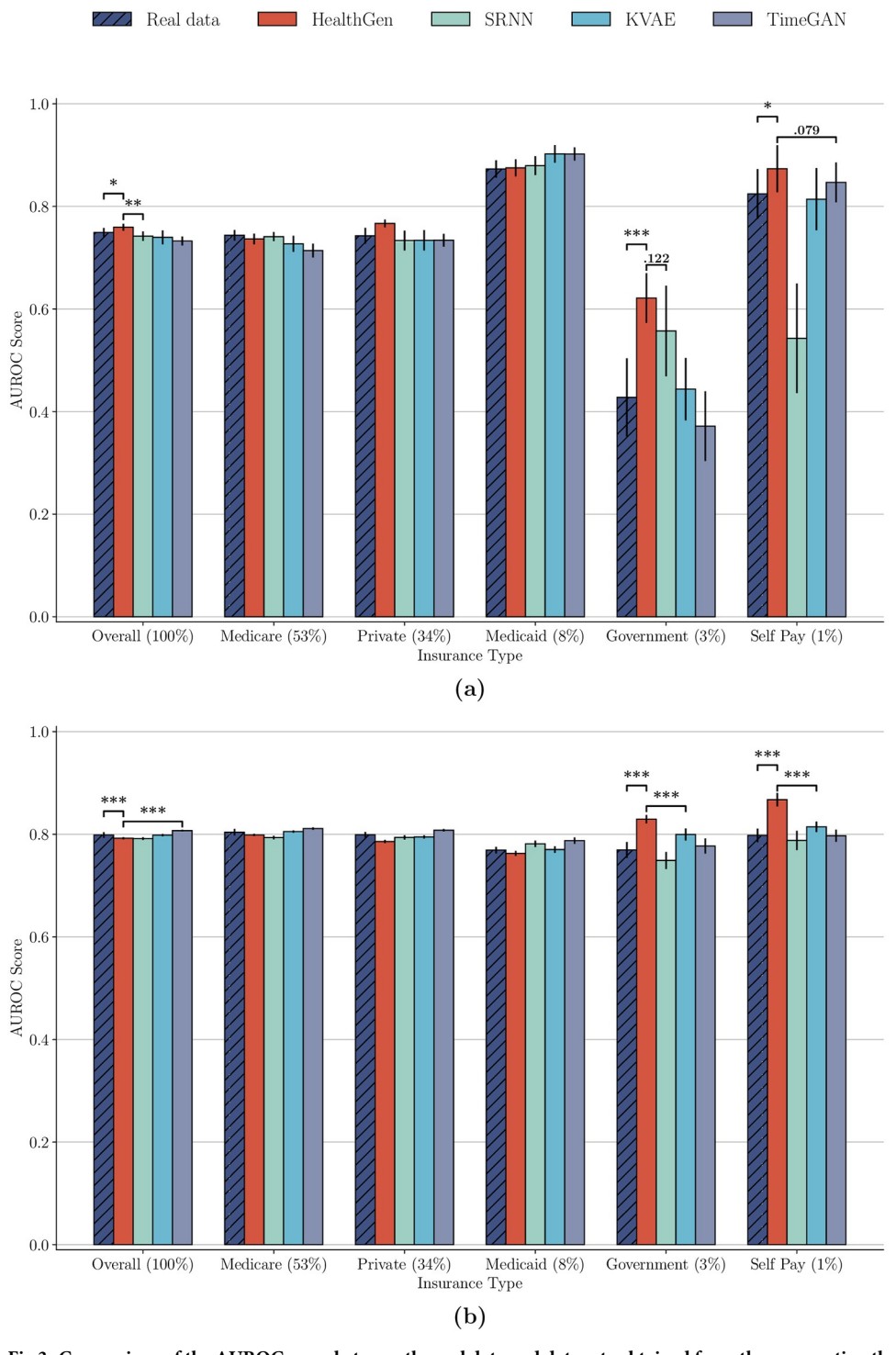

**Fig 3. Comparison of the AUROC score between the real data and data sets obtained from the augmenting the real data with the synthetic patients from the considered generative models.** We report the scores for each different insurance type on the (a) `colloid_bolus` as well as the (b) `vaso` classification task. Significance levels between groups of interest are shown with brackets, where * corresponds to $p < 0.05$, ** to $p < 0.01$ and *** to $p < 0.001$.

small. Interestingly, the performance of the Self Pay class, which is also heavily underrepresented, is also boosted, even if it was already at a high level to begin with. While some other baselines also manage to boost the score of the Government insured class, they either fail to do so to the same degree as our approach, or they also decrease the performance for another class. These findings are further underlined by the resulting fairness metric scores presented in Table D in S1 Appendix. Our model's superiority is further demonstrated by the fact that our conditional augmentation leads to a boosting of the overall score as well.

A second setting in which our model provides a benefit for underrepresented classes is the `vaso` task, again looking at different insurance types. Here, the performance on the minority groups of Government and Self Pay insured patients is not as dramatically lower compared to the other majority groups, but our approach to augment the real data still provides a significant benefit. Visualized in Fig 3(b), our augmentation boosts the performance of the downstream classifier for the two smallest classes significantly, even in a setting where their score is not severely below that of the larger groups to begin with.

## 2.5 Privacy

To qualitatively assess if our model simply memorizes the training data and reproduces it at generation time, we visualize time series of a randomly selected, synthetically generated sample and time series of the three closest samples (nearest neighbours) in the training data. In Fig 4, we compare the corresponding features of the synthetic and real samples side-by-side and observe that while certain patterns are shared, the synthetic data is not a copy of any of the real patients. This indicates that our model does not memorize the sensitive training data, allowing us to conjecture that it is privacy preserving to a certain degree, and sharing synthetically

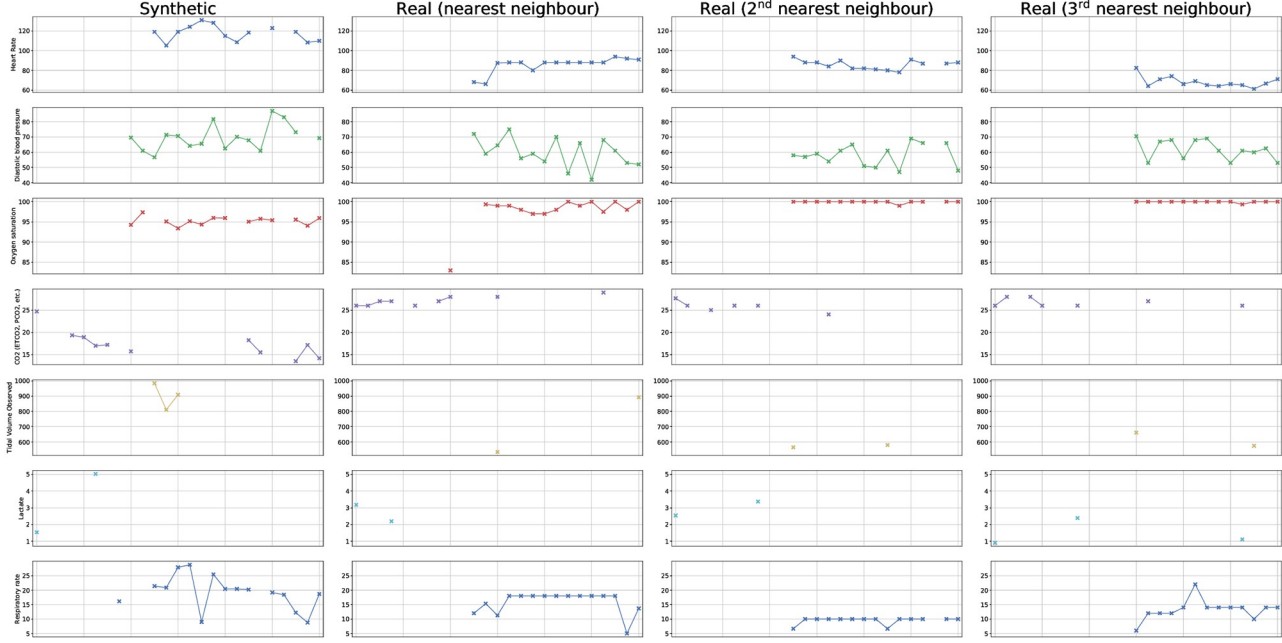

**Fig 4. Comparison of the time series of a randomly sampled, synthetically generated patient and the corresponding time series of the three closest real patients (nearest neighbours) in the training data.** While certain characteristics such as the number of missing values per feature or dynamics are similar between the synthetic sample and its nearest neighbours, we observe that the synthetically generated data is not a copy of the real data, indicating that our method does not memorize the data it sees during training. This experiment is merely meant to visually check if the generated data is an exact copy of the real data. The overall quality of the synthetic data is measured quantitatively by our evaluation scheme, not visually assessed here.

generated patient cohorts does not jeopardize the private information of the real patients our model was trained with. We stress that this visualization is only meant as a qualitative visual inspection that the synthetically generated data is not a copy of any real patients. This is *not* a metric of how well the synthetic data captures the relevant characteristics of the real data, as this is measured quantitatively via our evaluation pipeline, described in detail in Section 4.

## 3 Discussion

We presented HealthGen, a deep generative model capable of synthesizing realistic patient time series data, including informative patterns of missingness, that is faithful to data observed in real patient cohorts. To study the quality of the generated patient cohorts, we trained our generative model on the MIMIC-III data set, consisting of labeled ICU patient time series, to synthetically generate EHR data and evaluate the utility of the generated data on the clinically relevant downstream task of classifying patients' medical time series. In an experimental comparison against existing state-of-the-art models for time series generation, we explored multiple dimensions of the generative capability of our proposed approach: first, we synthesized patient cohorts with the same distribution of static variables as the real training data and observed that the data generated by our model is significantly more faithful to the real data across all evaluated downstream clinical prediction tasks than existing state-of-the-art approaches. In a second experiment, we demonstrated that HealthGen is capable of conditionally generating synthetic patient cohorts with static variable distributions that differ from the underlying, real data, without sacrificing the quality of the generated samples and boosting the fairness of the resulting synthetic patient cohorts. Finally, we identified settings where Health-Gen can alleviate issues of unfair downstream classification performance between demographic subpopulations that arise in the real data, by augmenting the real patient cohorts with more diverse, synthetic samples.

### 3.1 Generating synthetic patient cohorts

A key motivation behind synthetically generating medical time series is the lack of access to this type of data for the development of downstream tasks. Data-driven approaches to assist clinical practitioners in diagnostic or therapeutic tasks promise significant improvements to the quality of healthcare we can provide in the future, but without sufficient data, both in terms of amount and quality, their development is impeded. Clinical institutions that collect this type of data at large are reluctant to centralize and share it, raising the question of how access to useful training and development data may be ensured. One approach that has been brought forward recently is the idea of synthesizing patient cohorts, with the hope that these generated data sets can then be shared freely. In this scenario, only one model hast to be granted access to the sensitive real data, while the synthesized cohorts that are generated by the trained generative model can be freely shared with anyone in need of data for developing a downstream task. The primary requirement for this generated data is that it must adhere to the characteristics of the real data in such a way that it allows for the meaningful development of downstream models. These models are then deployed in the real world, to be used with real data. We demonstrate our model's capacity to fulfill precisely these requirements. On five different clinical downstream tasks, the synthetic patient cohorts generated by our approach are closer aligned to the real data, evident from their classification scores being closer to that of the downstream task trained on real data, than any competing baselines. While in no setting the generated data ever outperforms the real data in terms of classification score, we stress that this cannot be expected and more importantly this does not diminish the obtained results. In practice, one would only have access to synthetic data for development of these downstream

models, so the performance obtained by training on real data is only considered for model selection of the upstream generative model, by means of providing a point of reference for comparison.

Our approach to synthetically generate realistic and useful medical time series data outperforms competing state-of-the-art models for a number of reasons. We include inductive biases aligned with the healthcare domain, in the form of explicitly modelling missing data patterns and separating the generation of these missing values from the generation of observable clinical variables. Furthermore, our model's capability to capture the influence of static and demographic variables on the generated data and to condition on them during the generation of the synthetic data adds to the expressive capacity of our architecture. To the best of our knowledge, no other models to generate time series in the medical domain explicitly model missing data patterns, even though their importance and prevalence in clinical data is well known. Instead of cherry picking features with low missingness or downsampling the temporal resolution of the data to alleviate missing values, we can generate time series data that is faithful to the characteristics of realistically occurring EHR data. Not only do we outperform the competing baselines on all of the considered downstream tasks, but we do so in a much more realistic setting than previously presented in the literature. This is a notable contribution, as we validate and compare models to generate medical time series with real-world downstream tasks for healthcare applications, giving our findings significantly more weight for clinical practitioners concerned with an impact beyond exemplary academic settings.

## 3.2 Conditional generation

In the context of healthcare applications, providing fair diagnostic models is of high ethical importance and increasing the fairness of such tools can have a potentially large impact. If an approach works well for the majority of a cohort at the cost of neglecting a certain subclass this can lead to systematically worse treatment of patients belonging to this group. Even a small increase in the predictive quality of a diagnostic tool can mean that hundreds or thousands of patients receive a treatment better aligned with their needs.

In addition to our model being able to generate synthetic patient cohorts of high quality and usefulness, we can do so with a high fidelity of control over the composition of cohorts, without having to sacrifice the quality of the generated data. In settings where the real data displays significant differences in performance between different subpopulations, utilizing this conditioning capability provides a benefit and yields more fair synthetic data sets. This increased fairness is evident from the lower variance between subpopulations when utilizing conditioning, compared to the unconditionally generated data of the other approaches. Importantly, this increase in fairness does not come at the cost of diminishing the performance of certain populations, but rather through an increase in the score of previously sub-par groups, which is also evident in the increase in overall score, with respect to our model when we do not condition. While conditional generation never hurts overall, we cannot boost the score of any subpopulation in any arbitrary setting. For the conditioning to provide a benefit, the real data must display performance imbalances between subpopulations. When this imbalance is not present and all subgroups perform similarly, conditioning should not be expected to provide an additional benefit, as the differences in subpopulations are not relevant for their classification.

The fact that we can successfully condition the generative process of our model to synthesize patients with given features indicates HealthGen's capability to correctly capture meaningful dependencies between high-level, time invariant patient features and their influence on the resulting dynamics of the generated covariates. The key modelling choices that enable this level of conditioning are the fact that we introduce an additional static latent variable to

capture time invariant patient states, as well as the inference procedure by which our model learns the dependencies between this time-invariant latent variable and the dynamics of the sequential data we are interested in generating. Splitting the high-level patient features from the dynamics on an architectural level encourages our model to focus on learning these concepts separately. Independence however does not follow from this separation, as high-level patient states will dictate the evolution of dynamical variables over time, which we capture in the dependencies of the static latent variable on time-varying observations during inference and vice versa during generation.

### 3.3 Real data augmentation

We have shown that by leveraging our model's conditional generation capability, we can synthesize data sets which are significantly more fair in their representation of subpopulations of patients. Having demonstrated this in the setting where access to the real data is not given for the development of downstream models, therefore having to rely on fully synthetic data, the question arises if conditionally generated data can also provide a benefit when we *do* have access to the real data. In a scenario where access to a real data set is given during the development of such a clinical tool, we investigated if augmenting the real data with synthetic patients of specific, underrepresented subpopulations can help to develop a more fair downstream classifier.

After identifying settings where certain subpopulations display significantly lower classification performances than the majority groups, in our final experiment, we demonstrate our model's capacity to increase the fairness of these real data sets through augmentation with synthetically generated data. This underlines the usefulness of synthetic data generation for a data augmentation task, which is orthogonal to the original objective of our model, namely generation of fully synthetic patient cohorts. That we can boost the performance of downstream tasks using these mixed-modality data sets consisting of real as well as synthetic data speaks to our model's capability to generate time series that are true to the real data in their informative characteristics and opens up even more possibilities of utilizing synthetically generated data in relevant, real-world applications.

Here, the effect of our approach's explicit modelling of static variables of interest and the resulting capability to condition on these becomes even more evident. While other generative models can also boost the classification of individual underrepresented classes through unconditional generation, our model proves to have a decisive advantage. The baselines are bound to generate some examples of the minority classes during generation, but we can generate these with high fidelity in a targeted fashion. The resulting augmented data set that follows from our approach manages to boost the score of underrepresented groups, without sacrificing the previously good score of any other subpopulations, which cannot be said for the baselines against which we compare. While we can provide a benefit via augmentation with generated data in specific settings, this does not hold in general. This implies that we cannot simply hope to boost any subpopulations downstream classification performance by generating more samples of this class. Our findings suggest that two main conditions must be met for augmentation to provide a benefit: the gap between the classification score of the minority group and the other groups must exceed a certain magnitude and the other groups must display a minimum score overall, in order for the model to have informative enough examples to learn from, even if they belong to an other group than the one we are interested in generating.

### 3.4 Limitations

In this work, we do not provide any strong guarantees on the privacy preserving nature of the generated data sets. While it may be interesting to investigate and extend our model in the

future in terms of rigorous differential privacy-preserving guarantees [49], we argue that we still generate synthetic data that is privacy-preserving to certain degree. For example, we provide experimental evidence that our model does not simply memorize the training data and reproduce it to generate synthetic cohorts. Moreover, it has also been shown that training neural network architectures with stochastic gradient descent intrinsically guarantees a certain degree of privacy [50], the extent of which is however still an open research question.

Furthermore, the quality and diversity of the generated data which our model produces is limited by the real data with which it is trained. We cannot hope to generate samples of data which are too far out of the distribution of patients which the model has seen during training. Furthermore, biases related to factors which our model does not condition on are likely to be reproduced in the generated data, although this is a fundamental issue all machine learning models face. A possible solution to this could be the integration of HealthGen into a federated learning framework as an avenue of future development. The initial motivation to synthetically generate EHR data is the lack of large publicly available data sets, with those that are available being only representative of a specific patient cohort. Training a generative model on multiple cohorts in a privacy preserving, federated fashion has been proposed to increase the diversity of the generated data and further catalyze the development of personalized medicine [51, 52]. While there is no reason to believe our model could not be applied to other data, validating the presented results on an external data set may lend our findings additional weight. However, the lack of access to suitable data sets that can be readily used is the key limiting factor to doing so.

## 4 Materials and methods

Here, we present the methodological details of the experimental pipeline used in this study. As the raw data is incompatible for training, the custom preprocessing pipeline introduced in Section 4.1 is employed to arrive at the necessary format. This formatted data can then be used to train the HealthGen model, presented in Section 4.2. Once trained, the model is capable of generating synthetic data sets, whose quality and similarity to the real data is quantitatively evaluated using the evaluation pipeline described in Section 4.4.

### 4.1 Data set and preprocessing

In our experiments, we use the publicly available Medical Information Mart for Intensive Care (MIMIC-III) data set [8] as input. In its raw form, it consists of the deidentified electronic health records (EHRs) of 53,423 patients collected in the intensive care units (ICUs) of the Beth Israel Deaconess Medical Center in Boston, Massachusetts, USA between 2001 and 2012. It consists of multiple tables containing the data of its over 50,000 patients. A single patient's information is linked across tables through a unique patient ID, and time series data contains a time stamp to maintain the correct temporal ordering of measurements. In this form, the sequential data is not ordered and many of the raw measurements represent the same concept, but are redundantly recorded under different names.

As a first preprocessing step, we employ a slightly modified version of the `MIMIC-Extract` pipeline [53]. This yields a data set containing the ordered time series of measurements of each patient, static patient variables such as age, sex, ethnicity and insurance type, and a sequence of binary labels at each time step, indicating whether a certain medical intervention was active or not. We apply the standard cohort selection found in the literature [54–56]: the first ICU admission of adult patients (at least 15 years old), with a minimum duration of at least 12 hours, resulting in a total number of $N = 34472$ patients.

At this point, the time series data is still irregularly sampled and asynchronous across different features of the same patient. Given a sampling frequency, we look at the resulting window around each time step and either record the measurement, or indicate the absence of a measurement with a NaN (Not a Number) value. We then truncate all time series to have the same, fixed length. In our setting we choose a sampling frequency of 4 steps per hour and truncate the sequences to have a total duration of 12 hours.

From the observed feature sequences we additionally extract a sequence of binary masks $\mathbf{m}_{1:T}$ indicating where a value in $\mathbf{x}_{1:T}$ is missing:

$$
m_{t,d} = \begin{cases} 1, & \text{if } x_{t,d} \neq \text{NaN}, \\ 0, & \text{otherwise.} \end{cases} \tag{1}
$$

Finally, we standardize all (non-missing) numerical values of $\mathbf{x}_{1:T}$ to empirically have zero mean and unit variance along each dimension $d \in D$, and replace the NaN values in $\mathbf{x}_{1:T}$ with zeros.

To obtain a binary label for a patient, we split the 12-hour sequence into three sections: a 6-hour observation window followed by a 2-hour hold-out section and finally a 4-hour prediction window. The label is then extracted from the prediction window: if an intervention is active at any time in this section, the label is positive, otherwise it is negative. Drawing inspiration from Suresh et al. [57], this procedure aims to create a fair prediction of future interventions from observed data by minimizing information leakage. If there is no gap between observation and prediction, oftentimes the last step of the observation contains enough information for a meaningful prediction. We extract five binary labels corresponding to different types of clinical interventions in the ICU: vent refers to mechanical ventilation, vaso to the administration of vasopressor drugs, colloid_bolus and crystalloid_bolus refer to colloid and crystalloid fluid bolus administration, respectively, and niv denotes non-invasive ventilation. An overview of the prevalence of overall positive samples for each of these labels is presented in Table B in S3 Appendix. Table C in S3 Appendix provides a summary of the extracted static variables and the representation of each sub-cohort and Table A in S3 Appendix presents all extracted time-varying features together with selected statistics.

After preprocessing, each patient is represented by a time series of inputs $\mathbf{x}_{1:T} = \{\mathbf{x}_t \in \mathbb{R}^D\}_{t=1}^T$, a time series of missingness masks $\mathbf{m}_{1:T} = \{\mathbf{m}_t \in \{0,1\}^D\}_{t=1}^T$, where $D = 104$, a vector of static features $\mathbf{s} \in \mathbb{R}^M$, $M = 4$ and a set of binary outcome labels $\mathbf{y} \in \{0,1\}^L$, $L = 5$. The time series $\mathbf{x}_{1:T}$ and $\mathbf{m}_{1:T}$ cover 6 hours of measurements at a resolution of 15 minutes between steps, resulting in a sequence of length 25. The final data set $\mathcal{D} = \{\mathbf{x}_{1:T}^n, \mathbf{m}_{1:T}^n, \mathbf{s}^n, \mathbf{y}^n\}_{n=1}^N$ is then split into a training, validation and test set, stratified with respect to the labels $\mathbf{y}$.

In Fig 1 we visualize an exemplary set of time series of one patient. We can observe that some covariates such as the heart rate or oxygen saturation have many successive measurements and their evolution over time can be directly studied, while others such as $CO_2$ are missing values at the majority of the time steps and the signal of their dynamics is much sparser. This visualization also shows the two types of correlations evident in the sequential data. Firstly, values of variables may be correlated over time, as we can see from the evolutions of the heart rate and the diastolic blood pressure. Secondly, *when* values were measured may correlate as well, i.e. the patterns of missingness for different input variables can be related.

## 4.2 The HealthGen model

Here we introduce the main technical contribution of this work: the generative model we propose for the task of conditionally generating medical time series data, which we christen

**HealthGen**. The HealthGen model consists of a dynamical VAE-based architecture that allows for the generation of feature time series with informative missing values, conditioned on high-level static variables and binary labels.

Machine learning models can generally be categorized into discriminative or generative models. While discriminative models aim to learn decision boundaries in the data, generative models aim to learn the underlying distribution of the data. In contrast to discriminative models, generative models allow samples from the data distribution to be drawn, enabling the generation of synthetic data sets. A widely used family of generative models are variational autoencoders (VAEs) [46], which also constitute the basis of our HealthGen model. VAEs consist of an encoder, or inference network, which encodes the data to a lower dimensional latent representation, and a decoder, or generation network, which takes the latent representation and attempts to decode the data back to its original state. By pushing the reconstructed data to be close to the original observation, as well as imposing constraints on the structure of the latent space, VAE models can efficiently learn to model the underlying data-generating distribution.

The HealthGen model leverages an extension of the VAE framework to sequential data, where sequential latent variables $\mathbf{z}_{1:T}$ describe the dynamics of the observed data in the latent space. In HealthGen, we introduce an additional static latent variable $\mathbf{v}$ to capture the time-invariant characteristics of the data. In the remainder of this section, we present the generative and inference models of HealthGen. In S2 Appendix, we provide a more detailed model description including the functional forms of all the distributions, and all implementation details required to reproduce our results.

**Generative model.**   As discussed in the previous section, our data consists of a feature time series $\mathbf{x}_{1:T}$ representing the physiological state of a patient, a sequence of binary missingness masks $\mathbf{m}_{1:T}$ indicating when a value of $\mathbf{x}_{1:T}$ is observed and when it is missing, static observable variables $\mathbf{s}$, and labels $\mathbf{y}$. The pattern of missingness is notably not random, and highly informative, as preliminary experiments have shown (see S1 Appendix). This result is in line with the findings of Che et al. [43], who show that missing values in medical time series play a key role in downstream predictive tasks. Given their evident importance, we explicitly model the missingness masks $\mathbf{m}_{1:T}$ alongside the observed feature sequence $\mathbf{x}_{1:T}$.

The decoder network of the generative model aims to generate $\mathbf{x}_{1:T}$ and $\mathbf{m}_{1:T}$ from the latent variables $\mathbf{v}$ and $\mathbf{z}_{1:T}$. The generative process starts from the static latent variable $\mathbf{v}$ with a fixed unconditional prior $p(\mathbf{v}) = \mathcal{N}(\mathbf{v}; \mathbf{0}, \mathbf{I})$, where $\mathcal{N}(\mu, \sigma^2)$ is the Gaussian distribution with mean $\mu$ and variance $\sigma^2$. The observed features $\mathbf{x}_{1:T}$ and missingness masks $\mathbf{m}_{1:T}$ are then independently generated as follows.

First, the sequence of missingness masks $\mathbf{m}_{1:T}$ is generated from the static latent variable $\mathbf{v}$, conditioned on the static observable variable $\mathbf{s}$ and label $\mathbf{y}$. We do not model this as a dynamical process, but rather generate the entire sequence in one step, modelling its joint distribution with independent Bernoulli distributions:

$$p_{\theta_{\mathbf{m}}}(\mathbf{m}_{1:T} | \mathbf{v}, \mathbf{s}, \mathbf{y}) \quad = \prod_{t=1}^{T}\prod_{d=1}^{D} \text{Bernoulli}(m_{t,d}; \mu_{t,d}). \tag{2}$$

The subsequent generation of the observed features sequence $\mathbf{x}_{1:T}$ is based on the SRNN model [44], including ideas from the DSAE [58] and the conditional VAE [59] models. A transition model between timesteps in the latent space is learned, and at each timestep $t$ the latent variable $\mathbf{z}_t$ is decoded together with the internal RNN state $\mathbf{h}_t$, with additional conditioning on the static latent $\mathbf{v}$, the static features $\mathbf{s}$, and the labels $\mathbf{y}$, to generate $\mathbf{x}_t$. The generative

distribution for $\mathbf{x}_{1:T}$ is given by:

$$p_{\theta_{\mathbf{x}}}(\mathbf{x}_t|\mathbf{z}_t, \mathbf{h}_t, \mathbf{v}, \mathbf{s}, \mathbf{y}) = \mathcal{N}(\mathbf{x}_t; \boldsymbol{\mu}_{\theta_{\mathbf{x}}}(\mathbf{z}_t, \mathbf{h}_t, \mathbf{v}, \mathbf{s}, \mathbf{y}), \text{diag}\{\boldsymbol{\sigma}^2_{\theta_{\mathbf{x}}}(\mathbf{z}_t, \mathbf{h}_t, \mathbf{v}, \mathbf{s}, \mathbf{y})\}). \tag{3}$$

Finally, the joint distribution of all variables, conditional on the observed static features $\mathbf{s}$ and labels $\mathbf{y}$, is:

$$
\begin{aligned}
&p(\mathbf{x}_{1:T}, \mathbf{m}_{1:T}, \mathbf{z}_{1:T}, \mathbf{h}_{1:T}, \mathbf{v}|\mathbf{s}, \mathbf{y}) \\
&= p(\mathbf{v})p_{\theta_{\mathbf{m}}}(\mathbf{m}_{1:T}|\mathbf{v}, \mathbf{s}, \mathbf{y}) \\
&\quad \prod_{t=1}^{T} p_{\theta_{\mathbf{x}}}(\mathbf{x}_t|\mathbf{z}_t, \mathbf{h}_t, \mathbf{v}, \mathbf{s}, \mathbf{y})p_{\theta_{\mathbf{z}}}(\mathbf{z}_t|\mathbf{z}_{t-1}, \mathbf{h}_t)p_{\theta_{\mathbf{h}}}(\mathbf{h}_t|\mathbf{x}_{t-1}, \mathbf{h}_{t-1}.\mathbf{v}),
\end{aligned}
\tag{4}
$$

A graphical representation of the dependencies implied by this generative distribution is shown in Fig 5.

To generate synthetic data, we sample the features and missingness masks from the generative model $p(\mathbf{x}_{1:T}, \mathbf{m}_{1:T}, |\mathbf{s}, \mathbf{y})$ using ancestral sampling as described in Algorithm 1. In practice, the conditioning is implemented by concatenating the vectors $\mathbf{s}$ and $\mathbf{y}$, so to perform

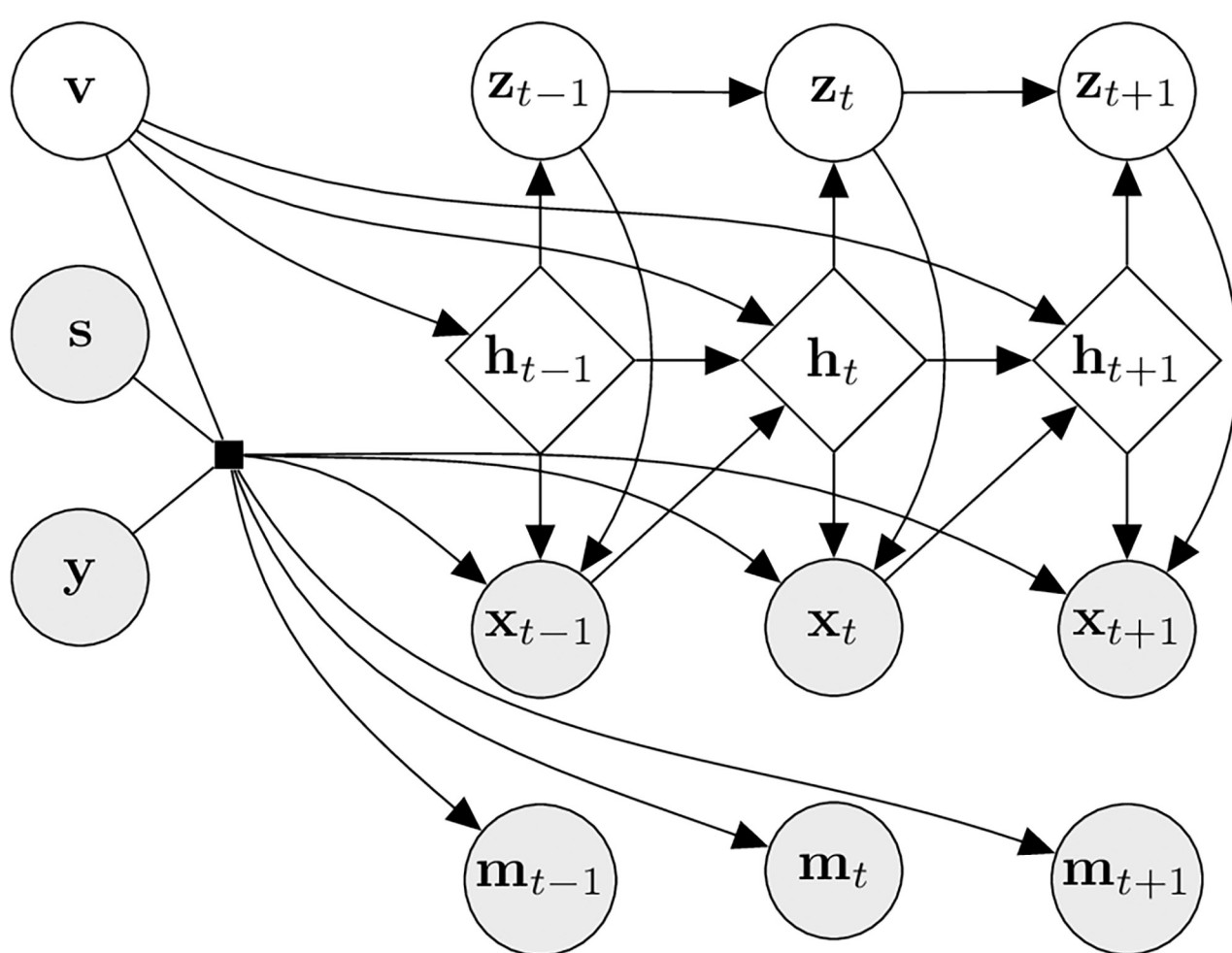

**Fig 5. Probabilistic graphical model of the generative process of HealthGen.** Diamond shaped nodes represent deterministic variables, round nodes probabilistic variables. Arrows represent direct dependencies. Shaded nodes represent observed or generated variables.

unconditional generation, we do not pass **s** at the conditioning step, but only **y**. Note that we sample $\mathbf{z}_0 \sim \mathcal{N}(\mathbf{z}_0; \mathbf{0}, \mathbf{I})$ rather than fixing it to $\mathbf{0}$, as we observed that this empirically yields synthetic data that is more useful for the downstream tasks.

**Algorithm 1** HealthGen sampling.

```
1: Set values for conditionals s, y
2: Sample v ∼ p(v) from static latent prior
3: Sample missingness masks m₁:T ∼ p_θₘ (m₁:T|v, s, y)
4: Sample z₀ ∼ 𝒩(z₀;0,I)
5: Initialize h₀ ← 0
6: for t ← 1 to T do
7:    Sample zₜ ∼ p_θ_z (zₜ|zₜ₋₁, hₜ)
8:    Sample xₜ ∼ p_θ_x (xₜ|zₜ, hₜ, v, s, y)
9:    Encode xₜ to obtain hₜ₊₁ ← eₕ(xₜ, v, hₜ)
10: end for
11: return x₁:T, m₁:T
```

**Inference model.**  Similarly to the generative process, we split the inference model into two steps, beginning with the inference of the static variable **v** from the observable feature sequence $\mathbf{x}_{1:T}$, the missingness pattern sequence $\mathbf{m}_{1:T}$ and the static features **s** as well as the label **y**. The approximate posterior distribution of **v**—the distribution of the static latent variable conditioned on the above mentioned dependencies—is subsequently formalized as:

$$q_{\phi_{\mathbf{v}}}(\mathbf{v}|\mathbf{x}_{1:T}, \mathbf{m}_{1:T}, \mathbf{s}, \mathbf{y}) = \mathcal{N}(\mathbf{v}; \boldsymbol{\mu}_{\phi_{\mathbf{v}}}(\mathbf{x}_{1:T}, \mathbf{m}_{1:T}, \mathbf{s}, \mathbf{y}), \mathrm{diag}\{\boldsymbol{\sigma}^2_{\phi_{\mathbf{v}}}(\mathbf{x}_{1:T}, \mathbf{m}_{1:T}, \mathbf{s}, \mathbf{y})\}). \tag{5}$$

The static latent variable **v** encodes the static features as well as the label, but it also encodes static information from the time series inputs. This allows our model to capture high-level information about a patient's state, that is not explicitly represented by any of the static features alone. By splitting the inference into a static latent variable and a latent time series representing the underlying dynamics of the observable features, our model learns to separate the time invariant content of a given sample from the dynamics that govern the evolution of the time-varying parts of its state. A patient's general state has a large effect on the temporal evolution of their time-varying lower-level states, which is reflected in our model's conditioning on the static features (both latent and observed) at multiple steps during the inference and generative processes.

This is formalized in the approximate posterior distribution of the latent sequence $\mathbf{z}_{1:T}$, which is defined as follows:

$$q_{\phi_{\mathbf{z}}}(\mathbf{z}_t|\mathbf{z}_{t-1}, \mathbf{g}_t) = \mathcal{N}(\mathbf{z}_t; \boldsymbol{\mu}_{\phi_{\mathbf{z}}}(\mathbf{z}_{t-1}, \mathbf{g}_t), \mathrm{diag}\{\boldsymbol{\sigma}^2_{\phi_{\mathbf{z}}}(\mathbf{z}_{t-1}, \mathbf{g}_t)\}) \text{ for } t > 1. \tag{6}$$

The overall inference model of HealthGen can then be written as follows:

$$q_{\phi}(\mathbf{z}_{1:T}, \mathbf{g}_{1:T}, \mathbf{h}_{1:T}, \mathbf{v}|\mathbf{x}_{1:T}, \mathbf{m}_{1:T}, \mathbf{s}, \mathbf{y}) \tag{7}$$

$$= q_{\phi_{\mathbf{v}}}(\mathbf{v}|\mathbf{x}_{1:T}, \mathbf{m}_{1:T}, \mathbf{s}, \mathbf{y}) \tag{8}$$

$$\prod_{t=1}^{T} q_{\phi_{\mathbf{z}}}(\mathbf{z}_t|\mathbf{z}_{t-1}, \mathbf{g}_t) q_{\phi_{\mathbf{g}}}(\mathbf{g}_t|\mathbf{x}_t, \mathbf{h}_t, \mathbf{g}_{t+1}, \mathbf{v}) p_{\theta_{\mathbf{h}}}(\mathbf{h}_t|\mathbf{x}_{t-1}, \mathbf{h}_{t-1}, \mathbf{v}). \tag{9}$$

A graphical overview of the described inference procedure, together with all modelled dependencies during the encoding step are visualized in Fig 6.

**Training HealthGen.**  VAE-based models are trained by optimizing a lower bound of the data log likelihood, which we adapt to optimize our model, as well. The general strucure of the ELBO includes a term which encourages a faithful reconstruction of the data after it has been

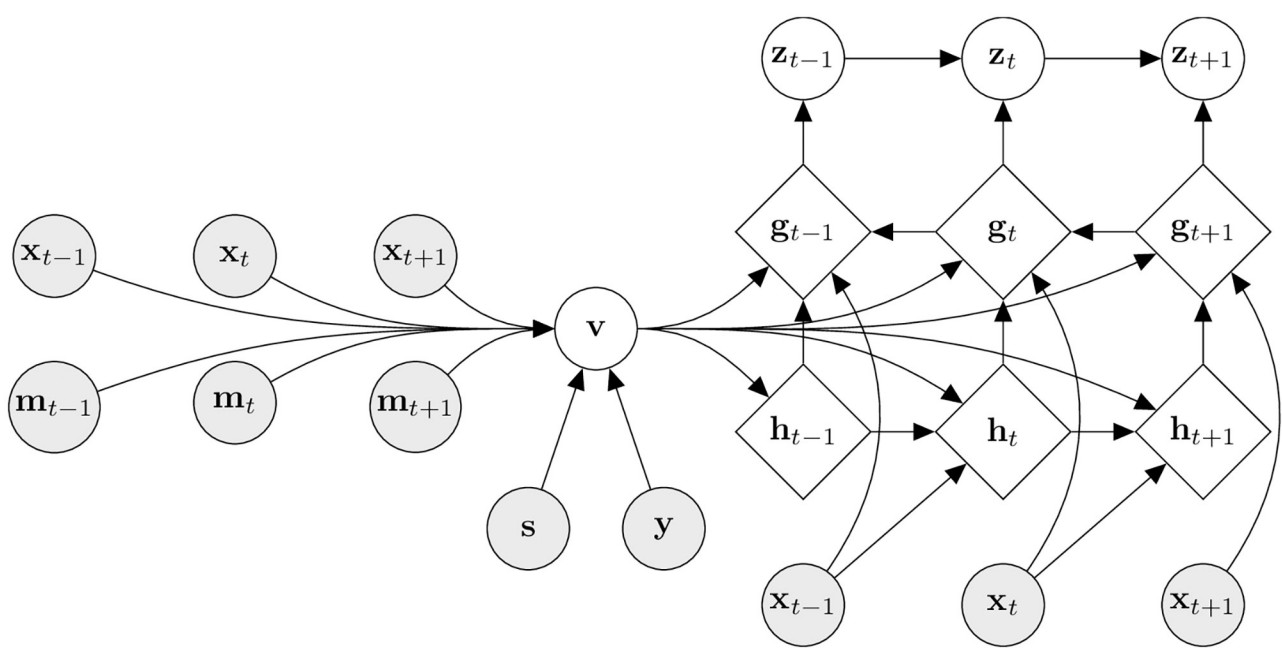

**Fig 6. Probabilistic graphical model of HealthGen at inference time.** Diamond shaped nodes represent deterministic variables, round nodes probabilistic variables. Arrows represent direct dependencies. Shaded nodes represent observed variables.

encoded and subsequently decoded, as well as a term penalizing the deviation of the latent distribution from a chosen prior distribution. HealthGen's parameters are trained by maximizing the Evidence Lower BOund (ELBO), a lower bound to the data log likelihood conditional on the labels and observable static variables. The final functional form of the ELBO which we optimize is given by:

$$
\begin{aligned}
\mathcal{L}(\theta, \phi) \quad &= \mathbb{E}_{q_{\phi_{\mathbf{v}}}(\mathbf{v}|\mathbf{x},\mathbf{m},\mathbf{s},\mathbf{y})}\left[\log p_{\theta_{\mathbf{m}}}(\mathbf{m}|\mathbf{v},\mathbf{s},\mathbf{y}) + \sum_t \mathbb{E}_{q_{\phi_{\mathbf{z}}}(\mathbf{z}_{1:t}|\tilde{\mathbf{g}}_{1:t})}[\log p_{\theta_{\mathbf{x}}}(\mathbf{x}_t|\mathbf{z}_t,\tilde{\mathbf{h}}_t,\mathbf{v},\mathbf{s},\mathbf{y})]\right] \\
&\quad - D_{\mathrm{KL}}(q_{\phi_{\mathbf{v}}}(\mathbf{v}|\mathbf{x},\mathbf{m},\mathbf{s},\mathbf{y})\|p(\mathbf{v})) \\
&\quad - \sum_t \mathbb{E}_{q_{\phi_{\mathbf{z}}}(\mathbf{z}_{1:t-1}|\tilde{\mathbf{g}}_{1:t-1})}[D_{\mathrm{KL}}(q_{\phi_{\mathbf{z}}}(\mathbf{z}_t|\mathbf{z}_{t-1},\tilde{\mathbf{g}}_t)\|p_{\theta_{\mathbf{z}}}(\mathbf{z}_t|\mathbf{z}_{t-1},\tilde{\mathbf{h}}_t))].
\end{aligned}
$$

We can discern a reconstruction term both for the observed feature sequence $\mathbf{x}_{1:T}$ and the missingess masks $\mathbf{m}_{1:T}$ in the first two summands, as well as KL divergences in the following terms penalizing the deviation of the inferred latent distributions of $\mathbf{v}$ and $\mathbf{z}_{1:T}$ from their respective priors. The full derivation used to arrive at this final objective function is presented in S2 Appendix. The parameters $\theta = [\theta_{\mathbf{x}}, \theta_{\mathbf{m}}, \theta_{\mathbf{z}}, \theta_{\mathbf{h}}]$ of the generative model and $\phi = [\phi_{\mathbf{v}}, \phi_{\mathbf{z}}, \phi_{\mathbf{g}}]$ of the inference model are jointly trained by descending on the negative gradient of the ELBO. The KL divergence terms have analytical expressions and all intractable expectations are approximated with Monte Carlo estimation. In practice, we mask the reconstruction loss term of the features $\mathbf{x}_{1:T}$ with the masks $\mathbf{m}_{1:T}$ to only take into account the learning signal of the features which have actually been observed.

### 4.3 Baseline models

We choose three generative models against which to compare our approach with: the SRNN [44], the KVAE [45] and the TimeGAN [47] models. These models were chosen in an effort to

select examples from the literature which represent the state-of-the-art in generative sequence modelling for different architectures. For technical details on the baseline models, please refer to the original papers.

The SRNN model is chosen to represent the "classical" dynamic VAE model: it utilizes RNNs as encoder and decoder and models the internal dynamics of the inferred latent sequence with an explicit transition model. In the comprehensive comparison between DVAE models provided by Girin et al. [60] it emerges as the most performative model, leading us to select it as the representative for this class of generative models.

The KalmanVAE is also included in our comparison due to its unique approach to model dynamics in the latent space. It combines a VAE with a classical linear state-space model to model the latent dynamics, resulting in interesting properties for the inference process. Fraccaro at al. [45] show that this approach works well in settings with well described dynamics, such as low dimensional mechanical systems, leading to the question of how well this translates to dynamics of clinical observables.

Models based on the GAN architecture have the reputation of shining when it comes to generating high quality synthetic data. To investigate if this is also the case in the setting we consider, we compare against the state-of-the-art GAN model for sequential data. In their original publication, Yoon et al. [47] also present one experiment on the MIMIC-III data set, making the TimeGAN model one of the most direct competitors to our approach a priori.

Since none of the models described above have the capability to generate data conditioned on labels $\mathbf{y}$, we provide a minor extension to all models, to enable a more fair comparison. Drawing inspiration from the Conditional VAE model [59], we repeat the labels $\mathbf{y}$ to all $T$ time steps $\mathbf{y}_{1:T}$ and encode them as an additional feature during training. The resulting latent sequence is then again extended by $\mathbf{y}_{1:T}$ before decoding. At generation time, we can choose $\mathbf{y}_{1:T}$ as we wish, append it to the sampled prior or random noise and decode to obtain a generated sample conditioned on the label we desire.

## 4.4 Evaluation

Quantitatively evaluating generative models is no trivial task, and in the setting where the generated data takes the form of real valued time series, this is especially true. Generative models that have been widely heralded as impressive examples of their class often convince the reader with generated human faces that are indiscernible from real images [61, 62]. In the medical setting, where specialized domain knowledge is necessary to tell the difference between real and fake samples, the quality of generated imaging data is presented to clinical experts, who then discriminate between synthetic and real samples [31].

Unfortunately, none of these approaches apply to the medical time series data we aim to synthesize. It may be possible to identify generated data with extremely low quality by visual inspection, but after a certain fidelity is achieved, discerning between a "better" or "worse" example of generated samples is no longer qualitatively possible.

To this end, we rely on the *Train on Synthetic, Test on Real* (TSTR) evaluation paradigm, first introduced by Esteban et al. [63]. A conceptual overview of our employed evaluation pipeline is presented in Fig 7. Let $\mathcal{E}$ denote the evaluation model trained on the real data $\mathcal{D}_{\text{train}}$ and $\hat{\mathcal{E}}$ denote the evaluation model trained on the synthetic data $\hat{\mathcal{D}}_{\text{train}}$. $\mathcal{E}$ and $\hat{\mathcal{E}}$ share the same architecture and are trained according to identical procedures with the same hyperparameters. Both models are then evaluated on the *same* held-out real test data $\mathcal{D}_{\text{test}}$:

$$e = \mathcal{E}(\mathcal{D}_{\text{test}}), \tag{10}$$

$$\hat{e} = \hat{\mathcal{E}}(\mathcal{D}_{\text{test}}). \tag{11}$$

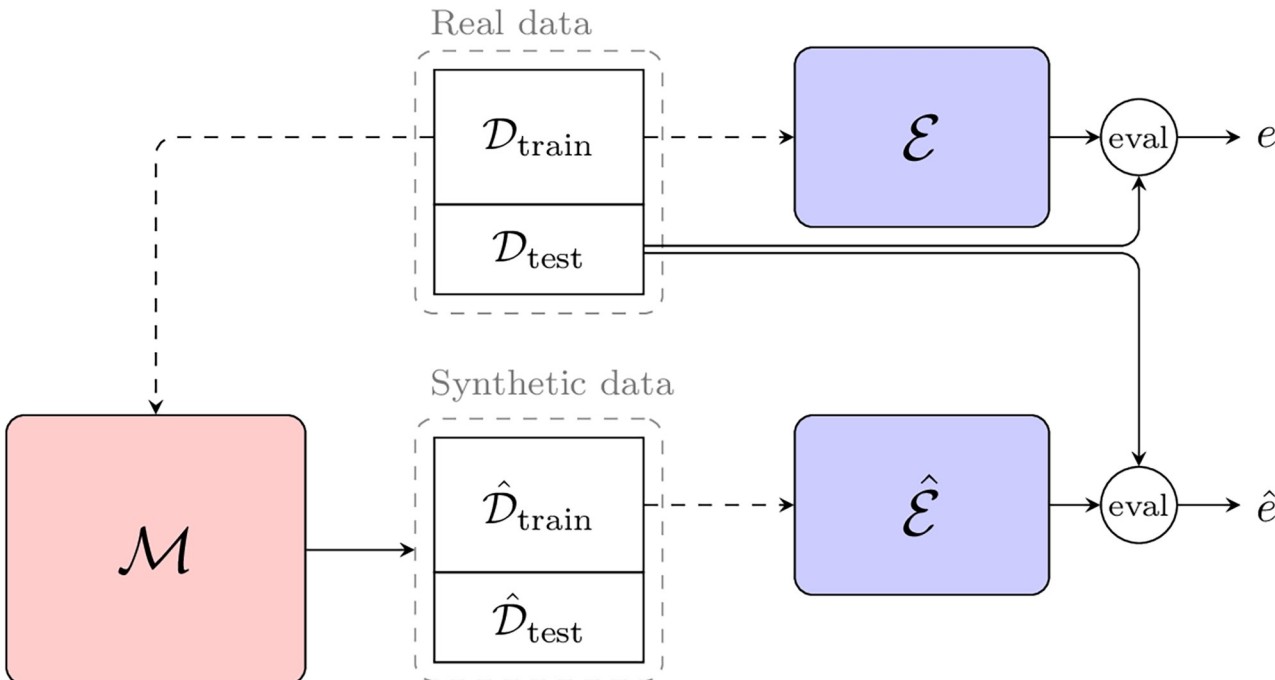

**Fig 7. Conceptual overview of the experimental pipeline.** The generative model $\mathcal{M}$ is trained with the real training data, allowing it to generate a synthetic data set. Two identical evaluation models $\mathcal{E}$ and $\hat{\mathcal{E}}$ are then trained with the real or synthetic data, respectively. Finally, these evaluation models are tested on the real data, yielding the metric $e$, derived from the real training set and that derived from the synthetic training data, $\hat{e}$.

The quantitative measure for how well a given generative model $\mathcal{M}$ performs is then measured in the difference between $e$ and $\hat{e}$. If the generative model $\mathcal{M}$ captures the dependencies in the real data that are informative for the downstream task represented by the evaluation model, and it is successful in synthesizing these in the generated data set $\hat{\mathcal{D}}$, this is reflected in a score $\hat{e}$ that is close, or ideally equal, to $e$.

The model that implements $\mathcal{E}$ in practice is the GRU-D model [43] for medical time series classification. Based on the Gated Recurrent Unit (GRU) [64], this model was specifically introduced for classifying time series with missing values in the medical domain. It identifies two characteristics of missing values in the healthcare setting: first, the value of the missing variable tends to decay to some default value if its last measurement happened a long time ago (homeostasis), and second, the importance of an input variable will decrease if it has not been observed for an extended period of time.

These principles are modelled with two separate decay mechanisms. If a variable is missing for a number of time steps, its value decays toward the empirical mean of its measurements over time. The second decay is applied to the internal hidden state of the RNN cell, to model the waning importance of states that have not been updated in a while. In addition to the input features $\mathbf{x}_{1:T}$, the GRU-D model also takes the masks $\mathbf{m}_{1:T}$ as well as the time since the last measurement $\boldsymbol{\delta}_{1:T} = \{\boldsymbol{\delta}_t \in \mathbb{R}^D\}_{t=1}^T$ as direct input.

While previous works have also used the TSTR framework to evaluate the quality of their generated medical time series data [47, 63], we argue that our setting is better suited to evaluate generative models in the healthcare domain. The key difference we wish to highlight is the proxy task, implemented by the downstream evaluation model, that is chosen for the evaluation. Past approaches have attempted to predict the value of the next step in the input sequence

[47], or to predict whether a time series surpasses a pre-defined threshold [63]. We opt for a downstream model that is specifically designed for a clinically relevant prediction task using real-world medical time series. This constitutes a setting much closer aligned to a real application in healthcare, and thus facilitating a comparison of generative models according to the relevant criteria instead of contrived metrics.

### 4.5 Uncertainty estimation

In all of our experiments, we repeat each run with five random initialisations of weights for the entire experimental pipeline, i.e. the generative model as well as the downstream evaluation model. For each generative model, we then choose the initialisation with the highest resulting downstream performance and estimate the 95% confidence interval of the mean of the AUROC score by performing bootstrap resampling 30 times on the resulting generated synthetic data set. This allows us to report and compare not only the obtained performance of the models we consider, but also the uncertainty of our chosen metric.

In the second and final experiments of this work, we additionally perform statistical tests to quantify the significance levels between competing approaches. Here, we take the scores of the bootstrapping for settings we wish to compare and perform the one-sided, parametric-free, Mann-Whitney U test [65], to determine the significance levels of competing approaches.

### 4.6 Memorization analysis

We analyze the privacy preserving characteristics of our model in similar fashion to DuMont Schütte et al. [31]. To find the nearest neighbour of a synthetic sample, among the real data, we measure the distances between their respective latent encoding. To this end, we take our trained model and encode a randomly sampled synthetic patient, yielding a 32-dimensional static latent vector $\mathbf{v}$ and a 32-dimensional latent time series $\mathbf{z}_{1:T}$ with 25 time steps. After flattening the time dimension in $\mathbf{z}_{1:T}$ and concatenating the static latent vector $\mathbf{v}$, we end up with an 832-dimensional latent representation of the synthetic patient. We repeat this process for all patients in the real training data set, again yielding an 832-dimensional latent representation for each real patient. Then, utilizing the cosine distance measure between vectors, we find the three nearest neighbours of the randomly sampled synthetic patient and plot the respective time series of this generated patient and its nearest neighbours amongst the training data in order to qualitatively compare them. A randomly sampled synthetic patient with its three nearest neighbours is visualized in Fig 4.

### Supporting information

**S1 Appendix. Preliminary findings and additional experimental results.**
(PDF)

**S2 Appendix. Model implementation and training details.**
(PDF)

**S3 Appendix. Data set characteristics.**
(PDF)

**S4 Appendix. Visualizations of synthetically generated data.**
(PDF)

## Author Contributions

**Conceptualization:** Simon Bing, Andrea Dittadi, Stefan Bauer, Patrick Schwab.

**Data curation:** Simon Bing.

**Formal analysis:** Simon Bing.

**Investigation:** Simon Bing.

**Methodology:** Simon Bing, Andrea Dittadi, Stefan Bauer, Patrick Schwab.

**Project administration:** Simon Bing.

**Software:** Simon Bing.

**Supervision:** Stefan Bauer, Patrick Schwab.

**Validation:** Simon Bing, Andrea Dittadi, Stefan Bauer, Patrick Schwab.

**Visualization:** Simon Bing.

**Writing – original draft:** Simon Bing, Andrea Dittadi.

**Writing – review & editing:** Simon Bing, Andrea Dittadi, Stefan Bauer, Patrick Schwab.

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
