## [Decision Letter · Decision Letter 0]

13 Apr 2022

PDIG-D-22-00047

Conditional Generation of Medical Time Series for Extrapolation to Underrepresented Populations

PLOS Digital Health

Dear Dr. Bing,

Thank you for submitting your manuscript to PLOS Digital Health. After careful consideration, we feel that it has merit but does not fully meet PLOS Digital Health's publication criteria as it currently stands. Therefore, we invite you to submit a revised version of the manuscript that addresses the points raised during the review process.

We look forward to receiving your revised manuscript.

Kind regards,

Gilles Guillot

Academic Editor

PLOS Digital Health

Journal Requirements:

1. Your co-authors, Stefan Bauer (stefan.a.bauer@gmail.com) and Patrick Schwab (patrick.x.schwab@gsk.com), have not confirmed authorship of the manuscript. We have resent them the authorship confirmation email; however please check that the above email address for them is correct and follow up personally to ensure they confirm. Please note that we cannot pass your manuscript to Production until we have received confirmations from all co-authors.

Just in case your co-authors are having difficulty confirming their authorship, you may advise them to send us an email at digitalhealth@plos.org and we will confirm their authorship on the authors' behalf.

2. Please update the completed 'Competing Interests' statement. Please declare all competing interests beginning with the statement “I have read the journal's policy and the authors of this manuscript have the following competing interests:”.

3. We ask that a manuscript source file is provided at Revision. Please upload your manuscript file as a .doc, .docx, .rtf or .tex. If you are providing a .tex file, please upload it under the item type ‘LaTeX Source File’ and leave your .pdf version as the item type ‘Manuscript’.

4. Please provide separate figure files in .tif or .eps format and remove any figures embedded in your manuscript file. Please also ensure that all files are under our size limit of 20MB. If you are using LaTeX, you do not need to remove embedded figures.

For more information about how to convert your figure files please see our guidelines: https://journals.plos.org/digitalhealth/s/figures

5. We notice that your supplementary figures and tables are included in the manuscript file. Please remove them and upload them with the file type 'Supporting Information'. Please ensure that all Supporting Information files are included correctly and that each one has a legend listed in the manuscript after the references list.

Additional Editor Comments (if provided):

This manuscript reports an important and interesting piece of work. 

In addition to the comments of three reviewers, I would add the following remark: most of the material presented is technical, abstract and based on highly specialised data analytics techniques. On the other hand, PLOS Digital Health readership is highly diverse and not necessarily familiar with the techniques implemented. There is a need to bridge the gap between the current state of the manuscript and the journal's readership. In particular, the Material and Method section does a poor job in its current state: it contains a lot of acronyms and abbreviations, it make uses of many variables and distributions that are not defined, the rationale of the model is not stated anywhere. Equations and diagrams can not be a substitute to explanations in plain English. This section requires a thorough re-writing.

Reviewers' comments:

Reviewer's Responses to Questions

**Comments to the Author**

1. Does this manuscript meet PLOS Digital Health’s publication criteria? Is the manuscript technically sound, and do the data support the conclusions? The manuscript must describe methodologically and ethically rigorous research with conclusions that are appropriately drawn based on the data presented.

Reviewer #1: Yes

Reviewer #2: Yes

Reviewer #3: Partly

2. Has the statistical analysis been performed appropriately and rigorously?

Reviewer #1: Yes

Reviewer #2: Yes

Reviewer #3: No

3. Have the authors made all data underlying the findings in their manuscript fully available (please refer to the Data Availability Statement at the start of the manuscript PDF file)?

Reviewer #1: Yes

Reviewer #2: Yes

Reviewer #3: Yes

4. Is the manuscript presented in an intelligible fashion and written in standard English?

Reviewer #1: Yes

Reviewer #2: Yes

Reviewer #3: Yes

5. Review Comments to the Author

Reviewer #1: The authors present an interesting method, HealthGen, to work on an important topic of generating medical time series (EHR) data for underrepresented population groups. The author report findings that suggest that compared with other methods including SRNN, KVAE, and TimeGAN, HealthGen performs better in generating realistic patient cohorts. In addition, the generated samples would lead to a better fairness representation for the minority group populations, and the trained model on these generated samples of underrepresented populations would have a better fairness than the model trained from the real dataset. Overall, this method and findings are important for model fairness and performance disparity reduction/elimination. The authors studied two population groups: ethnicity and insurance types, and showed that the performance from the method proposed in this manuscript worked best in most of scenarios. Overall, this method and findings are important, the manuscript is well written, the message is clear, and the organization is easy to follow. Still, the findings remain limited and the authors should consider the following comments: 

Minor: 

The reference should be sorted in order. Reference [1] is not cited in the article. 

Major: 

Since the author is working on underrepresented population groups, it would be better to discuss some other works including Federate Learning, transfer learning [1, 2], population allocation [3], meta learning [4], etc. Also, it would be better to mention some bias reduction and mitigation methods [5].

Another issue is the confounding factors [6,7]. It would be better to consider some factors, like age, sex, education background, ethnicity, etc. If you do not want to consider these confounding factors, please discuss the reasons. 

[1]: Wang, Z., Dai, Z., Póczos, B. and Carbonell, J., 2019. Characterizing and avoiding negative transfer. In Proceedings of the IEEE/CVF Conference on Computer Vision and Pattern Recognition (pp. 11293-11302). 

[2]: Gao, Y. and Cui, Y., 2020. Deep transfer learning for reducing health care disparities arising from biomedical data inequality. Nature communications, 11(1), pp.1-8.

[3]: Rolf, E., Worledge, T.T., Recht, B. and Jordan, M., 2021, July. Representation Matters: Assessing the Importance of Subgroup Allocations in Training Data. In International Conference on Machine Learning (pp. 9040-9051). PMLR. 

[4]: Qiu, Y.L., Zheng, H., Devos, A., Selby, H. and Gevaert, O., 2020. A meta-learning approach for genomic survival analysis. Nature communications, 11(1), pp.1-11.

[5]: Thompson, H.M., Sharma, B., Bhalla, S., Boley, R., McCluskey, C., Dligach, D., Churpek, M.M., Karnik, N.S. and Afshar, M., 2021. Bias and fairness assessment of a natural language processing opioid misuse classifier: detection and mitigation of electronic health record data disadvantages across racial subgroups. Journal of the American Medical Informatics Association, 28(11), pp.2393-2403.

[6]: Ewers, R.M. and Didham, R.K., 2006. Confounding factors in the detection of species responses to habitat fragmentation. Biological reviews, 81(1), pp.117-142.

[7]: Sul, J.H., Martin, L.S. and Eskin, E., 2018. Population structure in genetic studies: Confounding factors and mixed models. PLoS genetics, 14(12), p.e1007309.

Reviewer #2: This paper by Bing et al. presents an interesting new approach to the

generation of synthetic medical time series data. The proposed

HealthGen model is based on the foundation of a RNN-based deep

learning architecture that incorporates static values as conditioning

factors on Tom series data generation processes. The authors use

this conduct multiple experiments demonstrating the potential utility

of this approach for generation of synthetic data that might

ameliorate poor model performance for specific subgroups, leading to

more fair models.

This paper presents a novel and thoughtful approach at the

intersection of two key problems: appropriate generation of

high-fidelity data for construction of medical models and disparities

in clinical AI. The analyses provided demonstrate the utility of the

model for developing datasets that overcome differential model

performance across ethnicities and sources of payment, two key sources

of potential biases in medical AI models. 

The paper is generally convincing and the model seems highly

promising. There are some concerns that should be addressed to

clarify the presentation of the work and to strengthen the argument.

The fidelity of the time series models is central to the success of

the proposed HealthGen model Put simply, the synthetic time series

must be close enough to the original time series to be useful as

synthetic data without being simplistic duplications of the original

data. Figure 1, Figure 4, and Appendix D are used to support the

argument that the synthetic time series are close, but not too close

to the originals. However, as these results are purely qualitative,

it's hard to tell if they are are generalizable. This argument would be

much more compelling if the authors could propose an appropriate

measure of similarity and to demonstrate that this measure showed

appropriate characteristics over a large dataset.

This suggestion is made with full understanding that the appropriate

metric is far from obvious, as the goal would be to measure success in

hitting a "sweet spot" between complete lack of correlation and

memorization. Possibilities such as distance correlation/covariance

might be considered, but it is not clear that they would

suffice. Additional challenges include the need to ensure that 

synthesized time series exhibit clinically appropriate and feasible

behavior. For example, a synthetic dataset with high correlation to an

actual dataset might still be inappropriate if the synthetic data

exhibiting unrealistically large fluctuations. If a metric-based

approach is not feasible, another alternative might be an adversarial

approach - if an appropriately-trained classifier was not able to

successfully classify sequences as either being real or synthetic,

this might be seen as an argument for the success of the data

synthesis approach.

On another, perhaps more minor note, the discussion of the colloid

bollus augmentation results in Figures 2 and 3 seems to understate

the potential importance of those results. Tables C.2 and C.3 clearly

illustrate that the number of government-funded colloiud bolus

positive samples is very small, representing 0.95% (for colloid) and

3% (government funded). From the base of roughly 34,000 patients, this

amounts to approximately 10 patients at the intersection of these two

groups (assuming that the two factors are roughly independent). It's

not surprising that the initial results are so poor, but the

remarkable improvement on the augmented data is well-worth

emphasizing.

Regarding the presentation of the methods, the figures are useful and

the equations are helpful. However, the details are a bit dense,

reading as if presented for a NeurIPS audience, as opposed to the

presumably broader audience that might be reading PLoS Digital

Health. A slightly gentler introduction might be preferred for this

audience. 

A minor concern: the lack of external validation might be seen as a

weakness. Although there is no reason to believe that this method

would not work on a second data set, this validation should be

explored at some point in the future, and deserves mention in the

current manuscript.

Reviewer #3: The paper presents HealthGen, a conditional generative model for synthetic patient cohorts. In addition of synthetic data more faithful to real data than previous state-of-the-art models, their model also allows of conditional generation of specific patient populations, a feature missing in previous state-of-the-art models. This conditional generation allows them to specifically generate data for underrepresented groups and therefore reduce downstream AI models’ inequality in predictive power between over- and underrepresented groups.

Most parts of the paper are well-written and it is easy to follow the authors arguments for the most part. Some parts of the paper could be rewritten to better integrate in the overall story and parts of their evaluation could be improved to better demonstrate the points the authors want to make. The introduction is especially well-written and gives an excellent motivation for the problem. My main issue with the paper is that the authors do not provide enough data to sufficiently support their claims.

Issues regarding the performance of the model include the following:

1. The paper sets up 4 static variables and 5 classification tasks, but the authors only show results for a fraction of possible combinations of static variables and tasks. Additional information is needed to show that the model’s performance holds for the other combinations of static variables and classification tasks.

2. In figure 4 the authors show a synthetic patient and data of the 3 closest real patients.

While the data of the real patients looks similar, the synthetic data looks very different (different value ranges, different trends), leading to doubts whether their method of identifying nearest neighbors for synthetic patients is working. Additional statistics for i.e. a sample of 10 synthetic patients and the distance to their nearest neighbors, as well as the distance between the nearest neighbors would give more insights.

3. As described in 2., Figure 4 leads to doubts whether their method of identifying nearest neighbors works. The authors should therefore consider using another baseline approach like dynamic time warping to measure the distances between patient time series to verify that their method is working.

4. For results 2.4 they make an unfair comparison: “Since the baselines cannot generate data conditioned on static variables, we have them unconditionally generate the same number of overall samples that our model augments the real data with.”

As the baseline models can create any number of synthetic patients, they should create synthetic patients (and discard synthetic patients from overrepresented classes) until they have enough to create a balanced dataset. This shows that the model is able to create underrepresented groups with the same quality as the other approaches, while also showing that their conditional approach makes this much simpler and more targeted.

5. The downstream classification model’s strong performance on some underrepresented groups indicates that group membership might already carry information. Some initial information on how the medical interventions are distributed over group memberships would help put the results better in context.

6. P.6 “In all considered tasks, our approach significantly outperforms the state-of-the-art models in synthetically generating medical time series.” This is not true. See for example Table 1 (improvement for colloid_bolus likely not significant), of Fig.3b overall (performance worse than baseline, significant difference).

7. Very small underrepresented groups see a large increase in predictive power of the downstream model. This could be because the generative model learns to ‘overfit’ on 

Issues regarding the improved fairness by using synthetic data:

8. The authors define a way to measure fairness, and claim that their approach produces fairer results, but they never provide numerical evidence to support this

9. If the model overfits on underrepresented groups, it is also likely to reproduce bias found in the dataset. “Furthermore, the quality and diversity of the generated data which our model produces is limited by the real data with which it is trained” is not enough of a warning for this possibility

10. Ideally, one would use more than one dataset for evaluation to ensure that the model actually learns to generalize the properties of underrepresented groups and not overfits on patterns observed in the data. However, I concede that this is very difficult due to the limited availability of medical datasets and their limited compatibility.

Other issues with the introduction:

11. The term missingness is mentioned in the introduction (p.3). While in the later context it becomes clear what missingness means, but a short explanation would improve reading flow here.

Other issues with the results section:

12. The initial description of the dataset should explicitly mention that it contains only patients from one hospital, i.e. “[…] patients that spent time in the intensive care unit (ICU) in the Beth Israel hospital …”. Together with the preceding introduction, this can otherwise make it look like the dataset contains a variety of patients from different populations and the HealthGen model can be used out-of-the-box by anyone wishing to improve their dataset diversity.

13. For Fig. 1 and Fig. 4 it is not clear why / how these features were selected for display from the set of all features. Reducing number of features and number of patients so the image can fit horizontally would immensely improve readability.

14. The benefit of having Fig. 1 is unclear, the displayed synthetic timeseries look very different from the real one, only the patterns of missingness seem appropriate.

6. PLOS authors have the option to publish the peer review history of their article (what does this mean?). If published, this will include your full peer review and any attached files.

**Do you want your identity to be public for this peer review?** For information about this choice, including consent withdrawal, please see our Privacy Policy.

Reviewer #1: No

Reviewer #2: Yes: Harry Hochheiser

Reviewer #3: No

---

## [Decision Letter · Decision Letter 1]

10 Jun 2022

Conditional Generation of Medical Time Series for Extrapolation to Underrepresented Populations

PDIG-D-22-00047R1

Dear Mr Bing,

We are pleased to inform you that your manuscript 'Conditional Generation of Medical Time Series for Extrapolation to Underrepresented Populations' has been provisionally accepted for publication in PLOS Digital Health.

Best regards,

Gilles Guillot

Academic Editor

PLOS Digital Health

Reviewer Comments (if any, and for reference):

Reviewer's Responses to Questions

**Comments to the Author**

1. If the authors have adequately addressed your comments raised in a previous round of review and you feel that this manuscript is now acceptable for publication, you may indicate that here to bypass the “Comments to the Author” section, enter your conflict of interest statement in the “Confidential to Editor” section, and submit your "Accept" recommendation.

Reviewer #3: All comments have been addressed

2. Does this manuscript meet PLOS Digital Health’s publication criteria? Is the manuscript technically sound, and do the data support the conclusions? The manuscript must describe methodologically and ethically rigorous research with conclusions that are appropriately drawn based on the data presented.

Reviewer #3: (No Response)

3. Has the statistical analysis been performed appropriately and rigorously?

Reviewer #3: (No Response)

4. Have the authors made all data underlying the findings in their manuscript fully available (please refer to the Data Availability Statement at the start of the manuscript PDF file)?

Reviewer #3: (No Response)

5. Is the manuscript presented in an intelligible fashion and written in standard English?

Reviewer #3: (No Response)

6. Review Comments to the Author

Reviewer #3: (No Response)

7. PLOS authors have the option to publish the peer review history of their article (what does this mean?). If published, this will include your full peer review and any attached files.

**Do you want your identity to be public for this peer review?** For information about this choice, including consent withdrawal, please see our Privacy Policy.

Reviewer #3: No
